# Remote Estimation of Mangrove Aboveground Carbon Stock at the Species Level Using a Low-Cost Unmanned Aerial Vehicle System

**Zhen Li [1]**, **Qijie Zan [2,3]**, **Qiong Yang [3]**, **Dehuang Zhu [1]**, **Youjun Chen [1,4]** and **Shixiao Yu [1,\*]**

[1]    School of Life Sciences/Guangzhou Key Laboratory of Urban Landscape Dynamics, Sun Yat-sen University, Guangzhou 510275, China; lizh69@mail2.sysu.edu.cn (Z.L.); zhudh5@mail2.sysu.edu.cn (D.Z.); chenyj73@mail2.sysu.edu.cn (Y.C.)
[2]    College of Life Sciences and Oceanography, Shenzhen University, Shenzhen 518060, China; zanqijie@infobigdata.com
[3]    Guangdong Neilingding-Futian National Nature Reserve, Shenzhen 518040, China; yangking78@126.com
[4]    School of Agronomy and Bioscience, Dali University, Dali 671003, China
\*    Correspondence: lssysx@mail.sysu.edu.cn; Tel.: +86-20-3933-2980

**Abstract:** There is ongoing interest in developing remote sensing technology to map and monitor the spatial distribution and carbon stock of mangrove forests. Previous research has demonstrated that the relationship between remote sensing derived parameters and aboveground carbon (AGC) stock varies for different species types. However, the coarse spatial resolution of satellite images has restricted the estimated AGC accuracy, especially at the individual species level. Recently, the availability of unmanned aerial vehicles (UAVs) has provided an operationally efficient approach to map the distribution of species and accurately estimate AGC stock at a fine scale in mangrove areas. In this study, we estimated mangrove AGC in the core area of northern Shenzhen Bay, South China, using four kinds of variables, including species type, canopy height metrics, vegetation indices, and texture features, derived from a low-cost UAV system. Three machine-learning algorithm models, including Random Forest (RF), Support Vector Regression (SVR), and Artificial Neural Network (ANN), were compared in this study, where a 10-fold cross-validation was used to evaluate each model's effectiveness. The results showed that a model that used all four type of variables, which were based on the RF algorithm, provided better AGC estimates ($R^2$ = 0.81, relative RMSE (rRMSE) = 0.20, relative MAE (rMAE) = 0.14). The average predicted AGC from this model was 93.0 ± 24.3 Mg C ha$^{-1}$, and the total estimated AGC was 7903.2 Mg for the mangrove forests. The species-based model had better performance than the considered canopy-height-based model for AGC estimation, and mangrove species was the most important variable among all the considered input variables; the mean height (Hmean) the second most important variable. Additionally, the RF algorithms showed better performance in terms of mangrove AGC estimation than the SVR and ANN algorithms. Overall, a low-cost UAV system with a digital camera has the potential to enable satisfactory predictions of AGC in areas of homogenous mangrove forests.

**Keywords:** mangrove forests; aboveground carbon stocks (AGC); Unmanned Aerial Vehicles (UAV); high spatial resolution orthoimages; species type; canopy height model (CHM)

## 1. Introduction

Globally, mangrove forests have been shown to contain significant carbon (C) pools, where an average estimate of 1023 Mg C ha$^{-1}$ has been suggested for mangroves in the tropics [1,2]. Although the total worldwide area of mangrove forests is only 137,760 km$^2$, accounting for just 0.7% of the

total tropical forest area [3], mangrove forests contribute 10–15% of all coastal sediment carbon storage [4]. Unfortunately, mangrove forests are a key ecosystem that suffer from intense anthropogenic disturbances [5,6] and severe stress from global climate change [7]. As a result, 2% of global mangrove C was lost between 2000 and 2012, which is equivalent to a maximum potential of 316,996,250 t of $CO_2$ emissions [8]. As such, analyses of carbon reserves in mangrove ecosystems are of great value and interest with respect to climate change adaptation and mitigation strategies such as the United Nation's Reducing Emissions from Deforestation and Forest Degradation (REDD+) program [9]. Moreover, it is of great practical significance to help developing countries reduce deforestation and degradation rates, build capacity for conservation and sustainable forest management, and enhance forest C stock.

In South China, with its population boom and rapid economic development, the area of mangrove forests has decreased to only 22,025 ha, which is less than half of the mangrove area in the 1950s [10]. Since the 1980s, the Chinese government has launched a series of programs to restore and rehabilitate mangrove forests, where 34 natural mangrove conservation areas have been established to date [11]. Recently, an increased number of studies have focused on the restorative effects of mangrove forests through artificial planting and their temporal changes [12,13]. However, a comprehensive examination of the C stocks of mangrove forests and their recovery process is still lacking, and the effects of these efforts require scientific and systemic study. In an attempt to provide insights into the dynamics of mangrove recovery and address many of the mangrove restoration problems, estimating mangrove C stocks has become an issue of great interest to both researchers and governments.

Traditional inventory taking of the C stock of mangrove forests are usually lengthy and expensive because of their location in intertidal zones. Therefore, the estimation of mangrove C stocks by means of field measurements combined with remotely sensed data is considered to be an ideal, cost-effective method [14]. Remote sensing models for estimating mangrove aboveground biomass (AGB) and aboveground carbon stocks (AGC) have been established from different sensing data such as Landsat [15], Systeme Probatoire d'Observation de la Tarre (SPOT) [16], IKONOS [17], Advanced Land Observing Satellite (ALOS) [18], synthetic aperture radar (SAR) [19,20], and Worldview [21]. Because mangrove forests have monospecific assemblages with different physical community structures (e.g., the diameter at breast height (DBH) and density), it has been emphasized that vegetation types should be considered for accurate AGC estimation. Research conducted by Zhu et al. [21] has demonstrated that species type information obtained from WorldView-2 images can significantly improve biomass estimation accuracy. Chen et al. [22] integrated conifer species for biomass mapping with airborne light detection and ranging (LiDAR) and aerial photography, concluding that the incorporation of species types reduced the RMSE (root mean square error) by 10%. Chadwick [17] integrated LiDAR and IKONOS multispectral imagery to map red and black mangrove species and their biomasses.

With the recent development of unmanned aerial vehicle (UAV) technology, aerial images obtained with a digital camera mounted on a UAV have been widely used in small-scale forest inventory which has the benefits of low cost and high flexibility [23–25]. Very-high-resolution imagery derived from UAV system (UAVs) has the potential for identifying mangrove species [26]. Moreover, object-based approaches combined with high-resolution imagery have been frequently applied for mapping mangrove species in recent years [26,27]. For example, Wang et al. [28] showed that the object-based method had a better overall accuracy than the pixel-based method for demarcating artificial mangrove species communities using 0.5-m Pléiades-1 imagery. Additionally, photogrammetric imaging supported by the structure of motion (SfM) technique and dense image matching has become a very interesting tool to collect three-dimensional (3D) information of objects [29,30]. These methods, in combination with RGB (red-green-blue) spectral data, have already been explored in terms of retrieval of canopy height [31–33] and vegetation biomass estimations [34–37]. Nevertheless, only a few studies applying the low-cost UAVs with RGB spectral data for the estimation of the mangrove biomass [27]. For example, Otero et al. [36] used UAVs with digital camera for retrieving mangrove AGB in Matang Mangrove Forest Reserve. Navarro et al. [38] integrated UAV-based plot data with Sentinel-1 and Sentinel-2 to estimate mangrove AGB in Senegal.

Additionally, the consideration of texture information, which is ignored by most previous UAV studies, has been shown to significantly improve the accuracy of biomass estimation compared to the use of spectral information alone [39]. We believe that combining these variables, including species type, canopy height, texture features and vegetation indices (VIs), could greatly improve the accuracy of mangrove AGC estimates. In addition, the contribution and performance of these variables, especially species type and canopy height, during the estimation still need quantitative evaluation.

Remote-sensing modeling is an important method of biomass estimation, and the selected estimation methods may largely affect the subsequent results [40]. Previous studies that estimated mangrove biomasses applied traditional linear-regression algorithms [27]. However, linear regression methods are based on the assumption of linear relationships between biomass and predictors or independent variables, and thus, they may not provide satisfactory results due to the complex relationships between remote-sensing variables and biomass [41–43]. In this case, nonparametric and machine-learning algorithms (MLAs), such as artificial neural network (ANN), support vector regression (SVR), and random forest (RF), can deal with nonlinear relationships, learn from limited training data, and successfully solve classification problems that are difficult to distinguish; such approaches have been widely employed in forest biomass estimation [41,42,44–47]. Nevertheless, there is no single MLA that performs best for every study object and area [41,44–47], and a comparison of MLAs is highly desired, which will help us select the most appropriate model.

In this study, hundreds of images with very high resolution were collected to estimate the AGC of the mangrove forests in the Futian Mangrove National Nature Reserve using a low-cost UAV system with a digital camera. Based on the overlapping photographs and the SfM method, a high spatial resolution UAV orthoimage and a Digital Surface Model (DSM) were obtained. Then, four kinds of variables, including species type, canopy height, texture features and VIs derived from the UAV data, were used to estimate the mangrove AGC. More specifically, our study aimed to: (i) evaluate the importance of these four kinds of variables and select the most suitable; (ii) compare the full model with all variables, species-based model and canopy height model for mangrove AGC estimation; and (iii) examine which MLAs (ANN, RF, SVR) provided better AGC estimation performance of mangrove forests.

## 2. Materials

### 2.1. Study Site and Species

The mangrove forest in the northern Shenzhen Bay, which is representative of those in South China, significantly decreased in area during the 1970s and 1980s due to the urbanization of the Shenzhen Special Economic Zone [48]. In order to reverse this degradation, the government established the Futian Mangrove National Nature Reserve (FMNNR) in 1984 to protect the wetlands and restore the degraded mangroves while maintaining economic and social development [49] (Figure 1). The study site is characterized by a subtropical monsoonal climate, with an annual precipitation of 1700–1900 mm, a mean annual relative humidity of approximately 80%, and an annual average temperature of 22.4 °C [13]. The tides in Shenzhen Bay are semi-diurnal, with a spring tidal range of about 1.9 m [50].

Approximately 89 ha$^2$ of the conserved mangroves in the FMNNR [51] are distributed along 9 km of the coast from east to west. At the study site, there are four main native mangrove species belonging to three genera from three families, namely, *Kandelia obovata* Sheue, H.Y. Liu & J. Yong (Rhizophoraceae), *Avicennia marina* (Forsk.) Vierh. (Acanthaceae), *Acanthus ilicifolius* (L.) (Acanthaceae) and *Aegiceras corniculatum* (L.) Blanco (Myrsineaceae). Two pioneer species with a fast growth rate, *Sonneratia apetala* Buch.-Ham. (Lythraceae) and *Sonneratia caseolaris* (L.) Engl. (Lythraceae), were introduced to the study area via afforestation in 1993 [52]. They rapidly proliferated into low tidal zones and they established a dense population covering approximately 4 ha$^2$ [51]. Because *A. corniculatum* and *A. ilicfolius* are shrubs, we focused on the tree species *K. obovata*, *A. marina*, *S. apetala* and *S. caseolaris* with zonation of a single dominant layer in FMNNR.

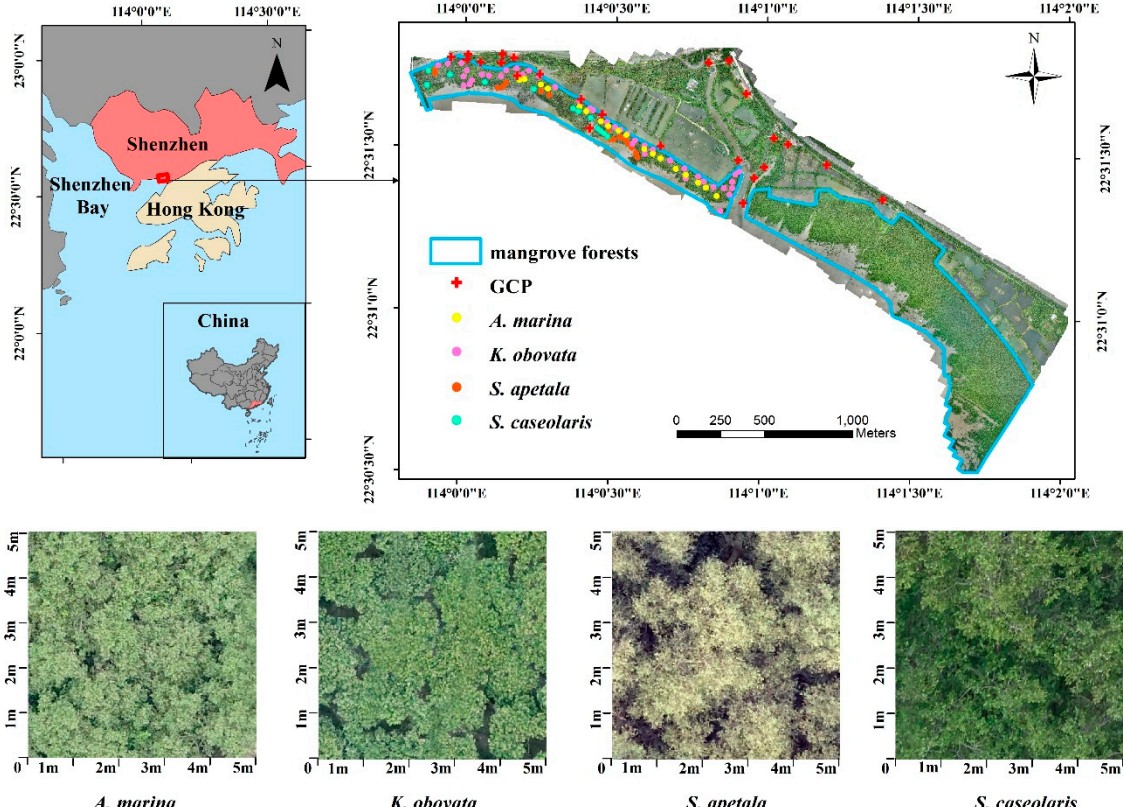

**Figure 1.** The location of the Futian Mangrove National Nature Reserve and the orthoimage produced from the unmanned aerial vehicles (UAV) flight in the study area with ground control points and field quadrats.

## 2.2. Field Measurements

Fieldwork was conducted in December 2017, where 88 quadrats of 5 m × 5 m with nearly mono-dominance (≥90% coverage of one species in the canopy) were established with an interval about 50~100 m between quadrats, and GPS coordinates were recorded at the center of each quadrat using a Real Time Kinematic GPS device (S86, SOUTH, China; see Figure 1). Because the eastern part of the reserve is a military frontier zone, which is off-limits to the public, the quadrats were located in the western part of the reserve, which included the four mangrove species of interest (Figure 1). All trees with heights ≥ 1.3 m in the quadrats had their heights and DBHs precisely measured following the standardized mangrove field protocol described by Kauffman and Donato [53]. All diameters were measured to the nearest 0.1 cm using a diameter tape. Heights were measured to a precision of 0.5 m for every tree using a handheld laser range finder (DISTO A3, Leica, Switzerland). In addition, the DBHs and heights of the standing dead trees were measured using the same methods used for the living trees [53]. As for the shrub, sapling of *A. ilicifolius* is the only dominant species in this layer, we also recorded its coverage and average height in each quadrat.

AGBs were determined through species specific allometric equations, where the height and DBH were considered as the independent variables for each tree species (Table 1). The climate conditions of the study area are similar to the sampled area of the studies below. Moreover, previously published studies gave a size range of the tree species included in this study, and our measurements were within these ranges, which allowed for the development of relevant equations for each species [54–57]. For standing dead trees, the aboveground biomass was estimated for each decay class according to Kauffman and Donato [53]. The C mass of the vegetation was calculated as the product of the vegetation biomass multiplied by the wood C concentration. C concentration was determined using results from previously published research of each tree species in South China (Table 1). For the biomass estimation

of *A. ilicifolius* sapling, we harvested all its saplings in ten 1 m × 1 m quadrats, and measure their dry mass as well as the carbon concentration (0.4225) of *A. ilicifolius* with the elemental analyzer in the lab [53]. An equation for estimating the biomass of *A. ilicifolius* was developed based on the height (biomass = 0.02775 H(m) − 0.003846, $R^2$ = 0.9225).

**Table 1.** Allometric equations and carbon content for calculating the aboveground carbon stacks of the mangrove species.

| Species | Allometric Equations | Carbon Concentration (%) | References |
|---------|---------------------|--------------------------|------------|
| *A. marina* | $\log B_{stem} = 0.544 \log(DBH^2 H) + 1.643$ | 41.2 | [54,55] |
| | $\log B_{branch} = 0.567 \log(DBH^2 H) + 1.897$ | 41.2 | |
| | $\log B_{leaf} = 0.287 \log(DBH^2 H) + 0.690$ | 39.8 | |
| *K. obovata* | $\log B_{stem} = 0.869 \log(DBH^2 H) + 2.162$ | 43.2 | [54,55] |
| | $\log B_{branch} = 1.253 \log(DBH^2 H) + 2.741$ | 43.2 | |
| | $\log B_{leaf} = 0.943 \log(DBH^2 H) + 1.706$ | 43.1 | |
| *S. caseolaris* | $\log B_{stem} = 0.807 \log(DBH^2 H) + 1.451$ | 43.2 | [55,56] |
| | $\log B_{branch} = 0.951 \log(DBH^2 H) + 0.321$ | 43.2 | |
| | $\log B_{leaf} = 0.931 \log(DBH^2 H) - 0.379$ | 39.9 | |
| *S. apetala* | $\log B_{stem} = 0.330 \log(DBH^2 H) - 0.959$ | 42.9 | [55,57] |
| | $\log B_{branch} = 0.388 \log(DBH^2 H) - 1.393$ | 42.9 | |
| | $\log B_{leaf} = 0.436 \log(DBH^2 H) - 2.500$ | 38.6 | |

DBH: diameter at breast height (m); H: height (m); $B_{AGB}$: aboveground biomass (kg); $B_{stem}$: stem biomass (kg); $B_{branch}$: branch biomass (kg); $B_{leaf}$: leaf biomass (kg).

## 2.3. Field-Based AGC

Table 2 shows the descriptive statistics of the quadrat-level AGC stock for the four mangrove assemblages in the FMNNR. Forest stand parameters (density, height and DBH) measured for individual trees within the quadrats were significantly different ($p < 0.05$) among the mangrove species. AGC values were calculated using the allometric equations in Table 1 and the C content of each species. The AGC ranged from 46.12 (Mg C ha$^{-1}$) to 153.12 (Mg C ha$^{-1}$), with the highest value calculated for *S. apetala*, followed by *K. obovata*, *S. caseolaris*, and *A. marina* (Table 2).

**Table 2.** Summary of the mangrove species information from the field sampling.

| Mangrove Assemblage | n | Density (trees·ha$^{-1}$) | Height (m) | DBH (cm) | AGC (Mg C ha$^{-1}$) |
|---------------------|---|--------------------------|------------|----------|----------------------|
| *A. marina* | 18 | 1555 ± 91 [a] | 6.03 ± 0.17 [a] | 16.56 ± 0.67 [a] | 46.12 ± 3.87 [a] |
| *K. obovata* | 42 | 7685 ± 679 [b] | 6.62 ± 0.13 [a] | 9.32 ± 0.35 [b] | 112.54 ± 7.98 [b] |
| *S. apetala* | 16 | 1625 ± 68 [a] | 8.82 ± 0.38 [b] | 17.48 ± 0.72 [a] | 153.12 ± 9.96 [c] |
| *S. caseolaris* | 12 | 1866 ± 114 [a] | 8.40 ± 0.32 [b] | 16.28 ± 0.66 [a] | 127.89 ± 7.27 [b] |

DBH stands for the tree diameter at breast height greater than 1.3 m; AGC: aboveground carbon stock. Data is given as a mean ± standard error. The mean values of density, height, DBH and AGC followed by different letters (a,b,c) within columns are significantly different at $p < 0.05$ based on Dunn's tests.

## 2.4. UAV Flight Data

The UAV flights were conducted in August 2017. A six rotor-wing UAV system (see Figure A1, ZR-66B, SOUTH, China) was used to collect the images. The UAV platform, which has a maximum flight time of 50 min under optimal weather conditions, was equipped with a Sony RX1RM2 camera and a GPS (SOUTH, China) as the payload. The camera has a focal length of 35 mm with 42.5 million effective pixels and produces images in three bands, namely, red (R, 625 nm), green (G, 550 nm), and blue (B, 485 nm). The flight mission was planned with the Mission Planer software that came with

the UAV (Figure A2). The flight altitude was set to 100 m above ground level. In total, 1997 valid images (7952 × 5304 pixels) were collected with 74% longitudinal overlap and 65% lateral overlap. Since the GPS loaded on the UAV platform can only provide rough positions, 24 ground control points (GCPs) were marked and positioned using a Real Time Kinematic GPS device (S86, SOUTH, China) with a horizontal accuracy of 0.01 m and an elevation accuracy of 0.02 m.

The professional software "Pix4Dmapper" (Pix4D, Lausanne, Switzerland), which has been widely used for UAV photogrammetric workflows, was used to reconstruct the study area using the UAV images. The processing procedure consisted of initial processing, point cloud and mesh, and DSM orthomosaic and index. 24 GCPs were used in the initial processing procedure. Subsequently, SfM dense point cloud data were produced after the point cloud and mesh procedure. Finally, two UAV raster products were derived from the images: an RGB orthoimage and the DSM. The ground sampling distance (GSD) was 0.02 m for the RGB orthoimage and 0.1 m for the DSM. The RMSE of the checkpoints was 4.2 cm along the *X*-axis (east), 5.6 cm along the *Y*-axis (north), and 8.7 cm along the *Z*-axis.

## 3. Methods

To estimate mangrove AGC, and quantify the effect of species type and canopy height metrics on the mangrove AGC estimation accuracy, we conducted three different experiments that involved building MLA models of the mangrove forest. In experiment 1, all the selected variables, including species type, canopy height metrics, VIs and texture features, were put into the models. In experiment 2, all variables except the canopy height metrics were put into the models. In experiment 3, all variables except the species type were put into the models.

First, we selected relevant variables for AGC estimation using the Boruta feature selection algorithm. Then, in each experiment, three MLAs were applied to develop the models, including RF, ANN, and SVM. After that, the performances of the nine models were assessed in terms of the coefficient of determination ($R^2$), RMSE, relative RMSE (rRMSE), mean absolute error (MAE), and the relative MAE (rMAE), all of which were based on 10-fold cross-validation. A flowchart of this study is shown in Figure 2.

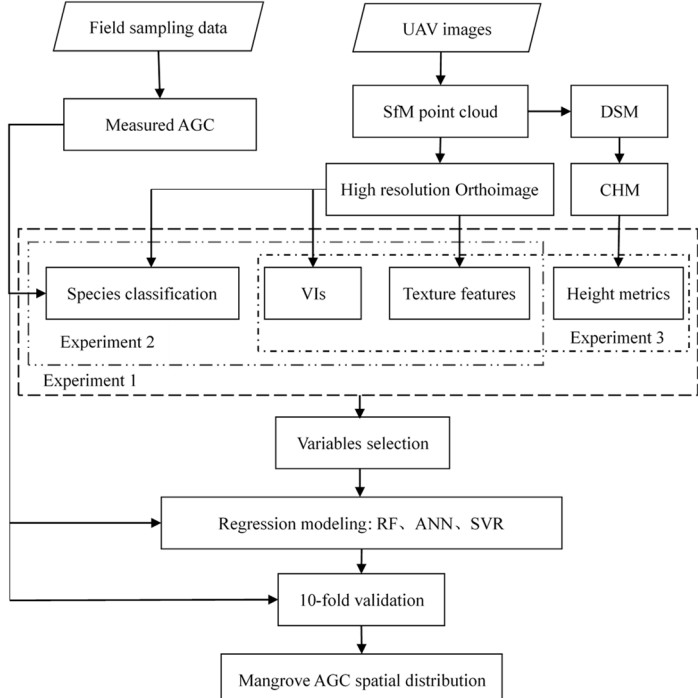

**Figure 2.** Flowchart of this study.

### 3.1. Classification of Mangrove Species

Objected-based approaches are based on segmentation, which divide the image into spatially continuous and spectrally homogeneous objects [16,26–28]. Image segmentation was processed in eCogniton Developer 9.0 (Trimble, Sunnyvale, CO, USA), using the multi-resolution segmentation (MRS) algorithm [26–28]. We set the segmentation parameters as follows: scale (50), color (0.9), shape (0.1), compactness (0.5), and smoothness (0.5). Three kinds of object features were also selected: spectral features (R/G/B/Brightness), geometry features (shape index and compactness), and texture features (Table 3).

We randomly selected 100 samples for each mangrove species based on artificial interpretation and field survey. In total 400 samples were divided into two sets randomly, with one set (200 samples) designated for training the Support Vector Machine (SVM) classifier [16,27] and the other for assessing the classification accuracy. The classification result was assessed by confusion matrix, which had an overall accuracy of 78% and a kappa coefficient of 0.73 (Table A1). The final distribution map of mangrove species is shown in Figure 3a.

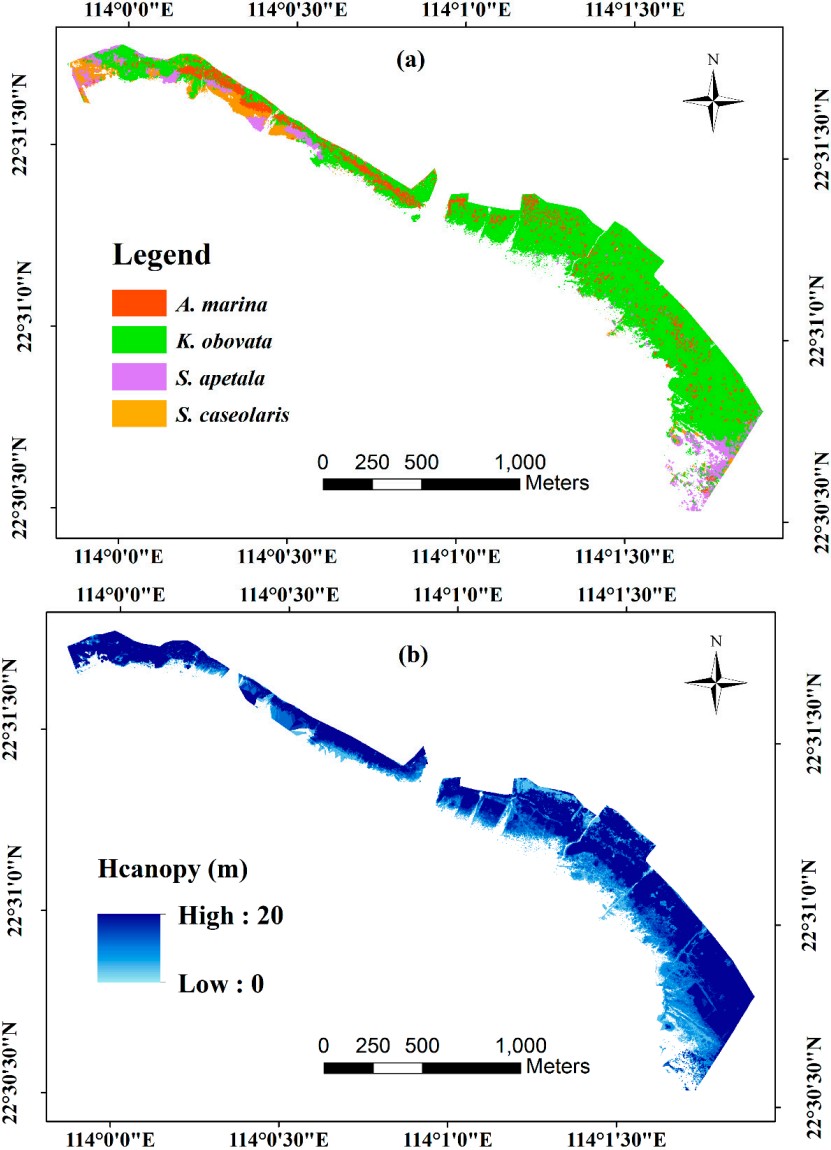

**Figure 3.** The classification of mangrove species (**a**) and canopy height model (CHM) (**b**) of the study area.

### 3.2. Predictor Variables Selection

#### 3.2.1. Calculation of the UAV Variables

To extract the height information of the mangrove forest, we produced a canopy height model (CHM) (0.1 m GSD) by normalizing the DSM to the average elevation of mudflat (Figure 3b). In this study, we calculated three CHM metrics, including the mean of the point height (Hmean), the standard deviation of the point height (Hstd), and the coefficient of point height variation (Hcv), according to previous research on canopy height estimation (Table 3) from CHMs [32,34,58].

12 selected VIs were calculated as different mathematical combinations of the RGB digital numbers from the orthoimage based on previous research on AGB and AGC estimations of UAV images (Table 3) [34,59,60].

**Table 3.** Definitions of variables used in this study.

| Variables | Definition | Source |
|---|---|---|
| | CHM metrics | |
| Hmean | $\text{Hmean} = \frac{1}{N}\sum_{n=1}^{N} h_n$ | |
| Hstd | $\text{Hstd} = \sqrt{\frac{1}{N}\sum_{n=1}^{N}(h_n - Hmean)^2}$ | |
| Hcv | $\text{Hcv} = \text{Hstd}/\text{Hmean}$ | |
| | Spectral vegetation index | |
| Red | Band1 (R) | |
| Green | Band2 (G) | |
| Blue | Band3 (B) | |
| Green-red ratio index (GRRI) | $\text{GRRI} = \text{G/R}$ | [62] |
| Green-blue ratio index (GBRI) | $\text{GBRI} = \text{G/B}$ | [63] |
| Red-blue ratio index (RBRI) | $\text{RBRI} = \text{R/B}$ | [59] |
| Normalized green-red difference index (NGRDI) | $\text{NGRDI} = (G - R)/(G + R)$ | [64] |
| Normalized green-blue difference index (NGBDI) | $\text{NGBDI} = (G - B)/(G + B)$ | [62] |
| Green leaf index (GLI) | $\text{GLI} = (2G - R - B)/(2G + R + B)$ | [65] |
| Visible atmospherically resistant index (VARI) | $\text{VARI} = (G - R)/(G + R - B)$ | [66] |
| Excess green index (EXG) | $\text{EXG} = 2G - R - B$ | [67] |
| Excess green minus excess red index (ExGR) | $\text{EXGR} = \text{EXG} - 1.4R - G$ | [34] |
| | GLCM texture measures | |
| Angular Second Moment (ASM) | $\text{ASM} = \sum_i \sum_j P[i,j]^2$ | |
| Contrast (Con) | $\text{Con} = \sum_i \sum_j P(i-j)^2 P[i,j]$ | |
| Correlation (Cor) | $\text{Cor} = \frac{\sum_i \sum_j ijP[i,j] - \mu_i\mu_j}{\sigma_i\sigma_j}$ | [61] |
| Entropy (Ent) | $\text{Ent} = -\sum_i \sum_j P[i,j]\ln P[i,j]$ | |
| Homogeneity (Hom) | $\text{Hom} = \sum_i \sum_j \frac{P[i,j]}{1+(i-j)^2}$ | |
| Mean | $\mu_i = \sum iP[i,j]$ | |
| Dissimilarity (Dis) | $\text{Dis} = \sum_i \sum_j P\left|i,j\right|$ | |
| Variance (Var) | $\sigma_i^2 = \sum i^2 P[i,j] - \mu_i^2$ | |

Note: $h_n$ = height of pixel in the CHM; N = number of pixels in the CHM; i is the row number of the orthoimage; j is the column number of the orthoimage; P[i, j] represents the relative frequency of two neighboring pixels.

In addition, we employed eight gray level co-occurrence matrices (GLCMs) to represent the texture measures (Table 3) [61]. The eight GLCM measures with eight different window sizes ($3 \times 3$, $5 \times 5$, $7 \times 7$, $9 \times 9$, $11 \times 11$, $13 \times 13$, $15 \times 15$, $17 \times 17$ and $19 \times 19$) were calculated from band 2 (Green) of the orthoimage in ENVI 5.3 (Exelis Visual Information Solutions, CO, USA).

To correspond to the field quadrats, 5 m × 5 m square vector buffers were set according to the GPS coordinates of the field sampling points. These pixel-wise UAV variables for each quadrat were averaged with a regional statistic tool (Zonal Statistics) to obtain the values per buffer in ArcMap10.2 (Esri, CA, USA).

### 3.2.2. Selection of Variables

To estimate the AGC of the mangrove forests, 88 variables, including species types, three height variables, 12 spectral variables, and 72 texture variables, were derived from the CHM and orthoimage as predictor variables. The variables were further analyzed with the Boruta feature selection algorithm, which is an all-relevant feature selection wrapper algorithm based on the RF algorithm [68]. Feature selection could be used to reduce the impact of a large number of explanatory variables. The Boruta method compares the importance of the original attributes with the randomly achievable importance, uses their replacement copies for estimation, and gradually eliminates irrelevant features to stabilize the test, thereby performing a top-down search for relevant features.

### 3.3. Model Regression and Accuracy Assesement

#### 3.3.1. Random Forest (RF)

The RF algorithm is an ensemble of many classification or regression trees that can reduce model overfitting [69,70]; a recent review of RF in remote sensing was given by Belgiu and Drăguţ [71]. Two parameters need to be defined in a RF: the number of trees to grow (ntree) and the number of variables to randomly sample as candidates at each split (mtry). Through tuning parameters, RF tries to maintain the prediction strength while inducing diversity among the trees [69]. In this study, mtry was set to the total number of available variables and ntree was set to 500 according to the review of [71]. The RF model was implemented with the "randomForest" package in R.

#### 3.3.2. Artificial Neural Network (ANN)

The ANN method simulates human brain learning processes via the establishment of linkages between input data and output data. Various ANN algorithms have been developed and applied in remote sensing; the reader is referred to the review by Mas and Flores [72] for more details. The back-propagation algorithm was used in this study. Two parameters need to be tuned: the number of units in the hidden layer and the weight decay. We examined a range of units, from 1 to 20, in the hidden layer, while the weight decay parameter ranged from 0.1 to 1. The other parameters were assigned default values. The optimal parameters were determined when the RMSE reached a minimum. The ANN model was implemented with the "nnet" package in R.

#### 3.3.3. Support Vector Machine (SVM)

The SVM method commonly uses a kernel function to transform training data into a high dimensional feature space, and to identify an optimal hyperplane that maximizes the distance between the hyperplane and the nearest positive and negative training examples [73]. A detailed review of SVM in remote sensing is provided by Mountrakis et al. [73]. In this study, a radial basis function (RBF) was used as the SVM kernel function. Two parameters need to be confirmed: cost and gamma. The "tune.svm" and "svm" functions were used to find the optimal combination of cost and gamma values. The SVM model was implemented with the "e1071" package in R.

3.3.4. Accuracy Assessment

A 10-fold cross-validation was used to assess the AGC estimates of the nine models. The cross-validation approach was based on the entire reference dataset, rather than using separate training and validation data subsets, which is a useful approach when only limited reference data is available [74]. Five validation measures of model performance were calculated from the 10-fold cross-validation including $R^2$, RMSE, rRMSE, MAE, and rMAE [37–43]. Models with higher $R^2$, smaller rRMSE and smaller rMAE values indicate a higher prediction accuracy:

$$R^2 = 1 - \frac{\sum_{i=1}^{n}(y_i - \hat{y}_i)^2}{\sum_{i=1}^{n}(y_i - \overline{y}_i)^2}$$

$$RMSE = \sqrt{\frac{\sum_{i=1}^{n}(y_i - \hat{y}_i)^2}{n}}$$

$$rRMSE = \frac{RMSE}{\overline{y}_i}$$

$$MAE = \frac{1}{n}\sum_{i=1}^{n}|y_i - \hat{y}_i|$$

$$rMAE = \frac{MAE}{\overline{y}_i}$$

where $y_i$ is the AGC measured in the field, $\overline{y}_i$ is the average value of $y_i$, $\hat{y}_i$ is the model-predicted value of AGC, and $i$ is the total number of matched quadrats.

In the 10-fold cross validation, each model produces AGC predictions for all the corresponding field measurements, while the regressions between the field measurements and model predictions were also conducted to further evaluate each model's performance.

## 4. Results

### 4.1. Variable Selection and Importance

Figure 4 presents scatterplots between three kinds of variable (texture features, VIs, and height metrics) and the measured AGC values of different species. Because texture features show high correlation with window size, we chose a window size of 19 × 19 to plot. As Figure 4 shows, the spectral signals, such as Green leaf index (GLI), Visible atmospherically resistant index (VARI), Green blue ratio index (GBRI), Normalized green-blue difference index (NGBDI), Normalized green-red difference index (NGRDI), Red-blue ratio index (RBRI), and Green-red ratio index (GRRI), of the UAV orthoimage tended to saturate with AGC in mangrove forests with such dense canopies.

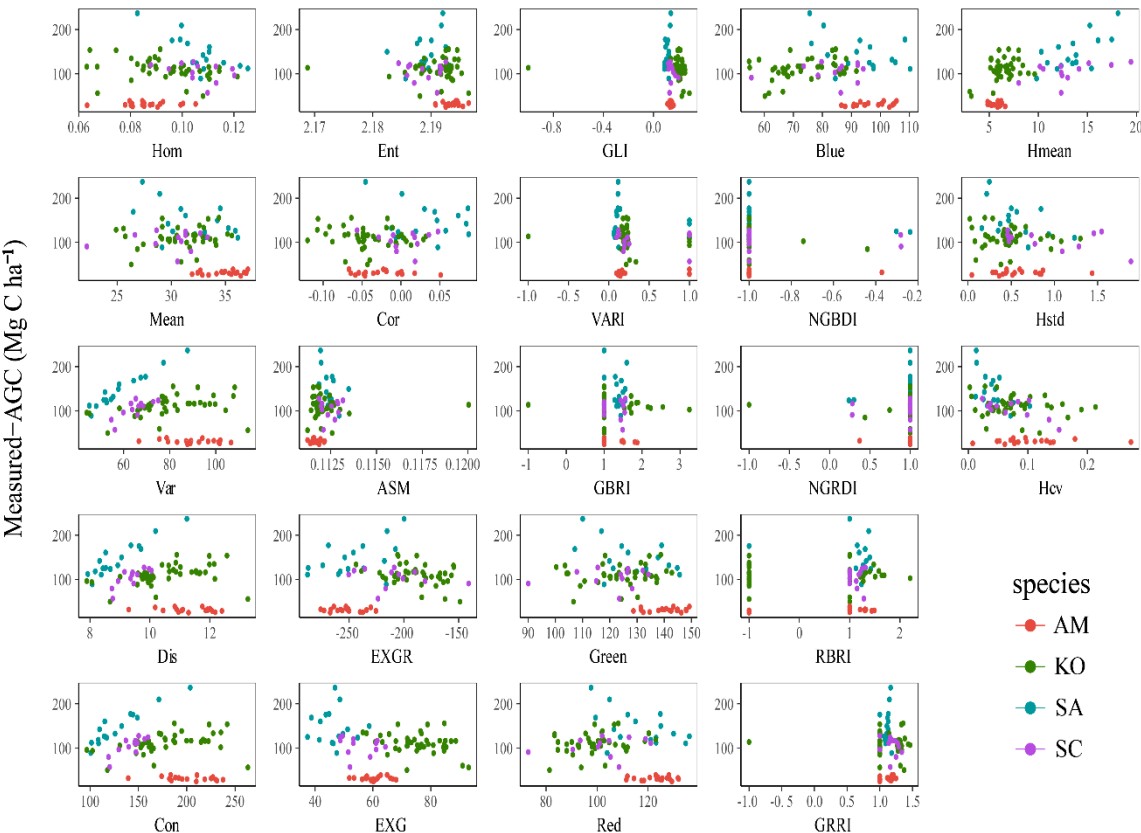

**Figure 4.** Scatterplots between the variables (texture features (19 × 19), vegetation indices and height metrics in Table 3) and the measured aboveground carbon stock (AGC) values of various species.

Figure 5a shows the 30 selected variables (from the 88 input variables) used for the AGC estimation, as determined with the Boruta method. The %IncMSE is the increase in the mean square error (MSE) of the predictions, which represents each variable's importance value in the RF model. In Experiment 1, 30 selected variables were put into the RF model, where the most important variable was species followed by Hmean, Excess green index (EXG), and Variance19 (Var19) (Figure 5a). In Experiment 2, 28 selected variables, excluding Hmean and Hcv, were put into the RF model, where species was most important variable, followed by EXG, Var19, and Variance13 (Var13) (Figure 5b). In Experiment 3, 29 selected variables, excluding species, were put into the RF model, where Hmean was the most important variable, followed by EXG, Excess green minus excess red index (EXGR), and Blue (Figure 5c). The selected variables were then used to build the ANN and SVR models for mangrove AGC estimation in each experiment.

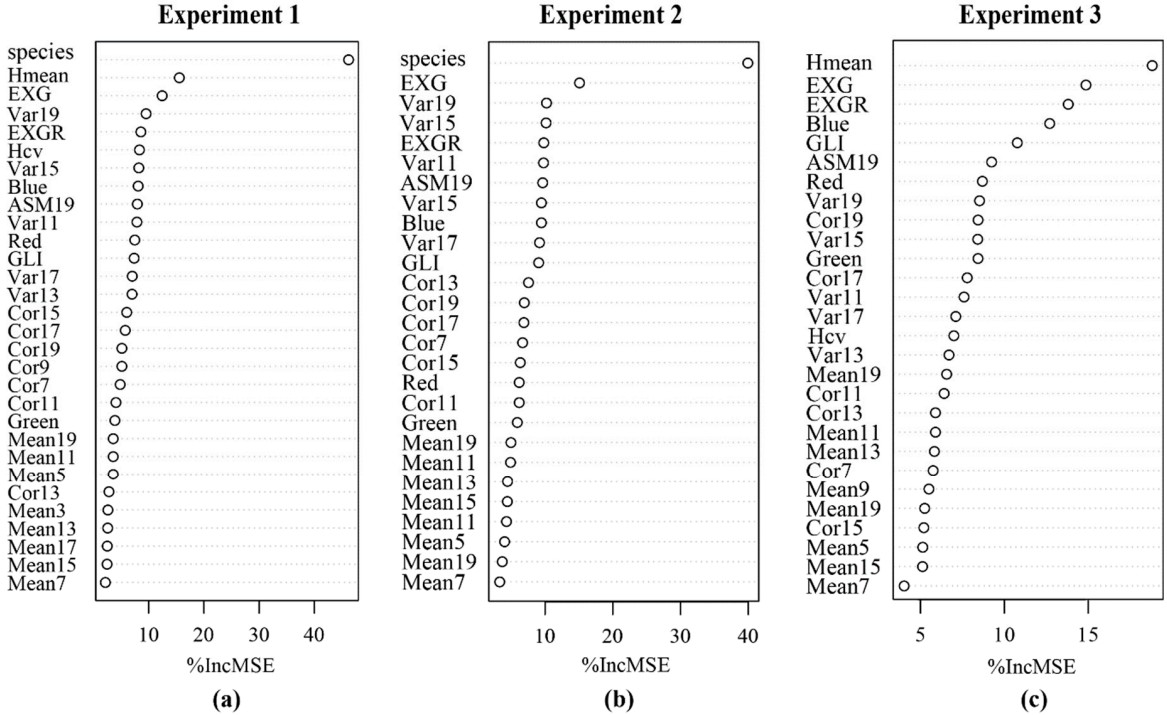

**Figure 5.** Selected variables of each experiment and their importance values (**a**–**c**).

## 4.2. MLA Models and Accuracy Assessment

In Experiment 1, four types of variables were put into the three models, where species was the most important variable (Figure 5a). The RF model achieved better accuracy than the other two models in terms of AGC estimation (Table 4). The SVR model and the ANN model also showed good performances with no statistically significant differences in terms of RMSE and MAE. In Experiment 2, three kinds of variables, excluding height metrics, were put into the three models, where species was also the most important variable (Figure 5b). The RF model also achieved the most accurate AGC estimation among the three models, while the ANN model was the least accurate (Table 4).

**Table 4.** Model performance for AGC estimation based on 10-fold cross validation. (RMSE: root mean square error; rRMSE: relative RMSE; MAE: mean absolute error; rMAE: relative MAE.)

| Model | $R^2$ | RMSE (Mg C ha$^{-1}$) | MAE (Mg C ha$^{-1}$) | rRMSE | rMAE |
|---|---|---|---|---|---|
| **Experiment 1:** all selected variables | | | | | |
| RF (a) | 0.81 | 20.46 [c] | 14.82 [c] | 0.20 | 0.14 |
| ANN (b) | 0.75 | 23.13 [bc] | 18.33 [bc] | 0.23 | 0.18 |
| SVR (c) | 0.80 | 21.21 [bc] | 16.82 [bc] | 0.21 | 0.16 |
| **Experiment 2:** selected variables without canopy height metrics | | | | | |
| RF (d) | 0.75 | 21.78 [c] | 15.88 [c] | 0.21 | 0.15 |
| ANN (e) | 0.64 | 31.34 [ab] | 26.67 [ab] | 0.31 | 0.26 |
| SVR (f) | 0.75 | 22.24 [bc] | 16.25 [bc] | 0.22 | 0.16 |
| **Experiment 3:** selected variables without species | | | | | |
| RF (g) | 0.65 | 27.32 [abc] | 21.78 [abc] | 0.27 | 0.21 |
| ANN (h) | 0.44 | 37.79 [a] | 31.23 [a] | 0.37 | 0.31 |
| SVR (i) | 0.66 | 26.39 [abc] | 21.54 [abc] | 0.26 | 0.21 |

The means of RMSE values followed by letters within columns are significantly different at $p < 0.05$ based on Dunn's tests. RF: Random Forest; ANN: Artificial Neural Network; SVR: Support Vector Regression

Overall, as shown in Table 4, the RF model in Experiment 1 showed the best accuracy and successfully explained 81% of the variance of the field-measured AGC values, with rRMSE = 0.20 and rMAE = 0.14. The models in Experiment 1 showed the best performance among the three experiments; this meant the four variable types considered in the models, including species, height metrics, Vis, and texture features, were suitable for AGC estimation. In addition, by comparing Experiment 2 with Experiment 3, and the importance values of the variables in Figure 5a, species was more important than height metrics. As for the model comparison, the RF models had better performance than the SVR and ANN models. This indicated that the RF algorithm was more useful and robust than the SVR and ANN algorithms for AGC estimation. The correlation between field-measured AGC and predicted AGC from the regression models are displayed in Figure 6.

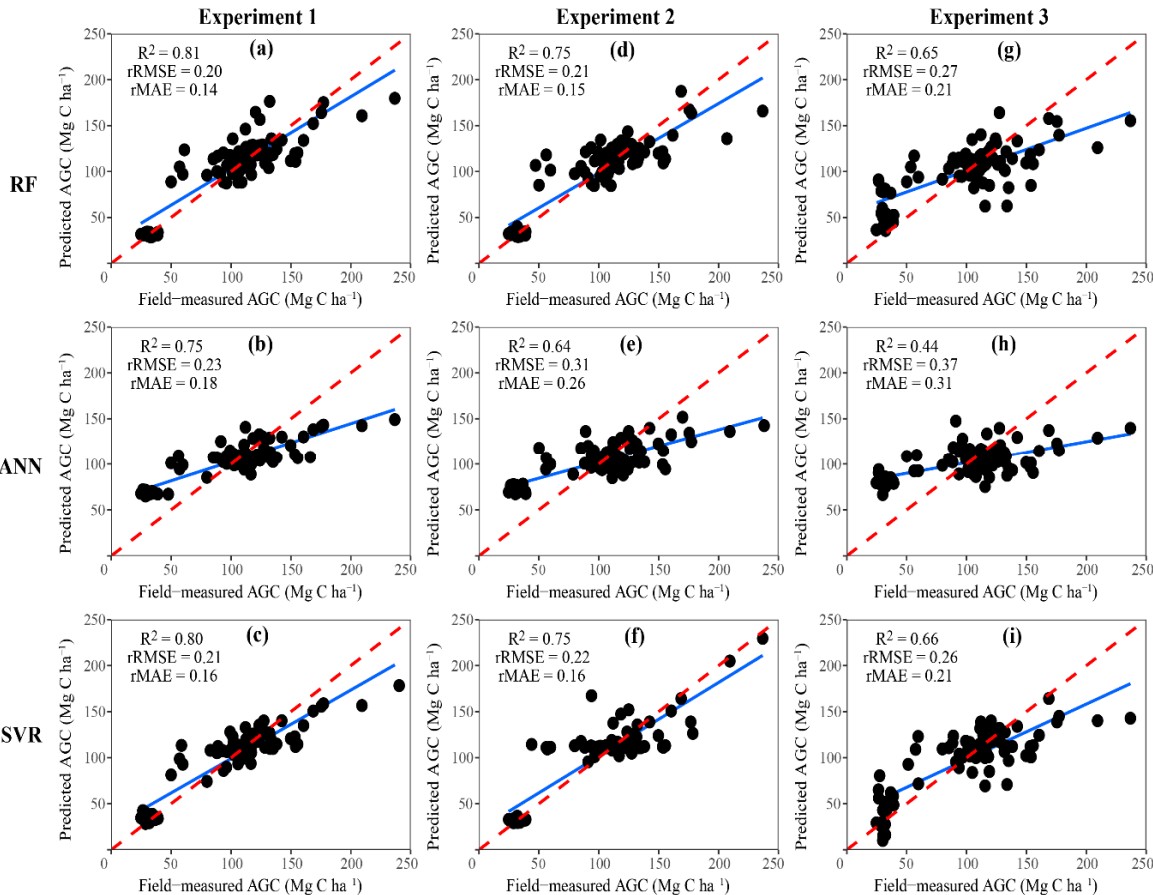

**Figure 6.** Field-measured aboveground carbon stock (AGC) values versus predicted AGC values from different models. Panels (**a**–**i**) are in consistent with Table 4.

### 4.3. Spatial Distribution of Mangrove AGC

Finally, an AGC map of mangrove forests in the study area (Figure 7) was produced based on the RF model from Experiment 1 (Table 4) and the 30 selected variables shown in Figure 5a ($R^2$ = 0.81, rRMSE = 0.20 and rMAE = 0.14). The estimated AGC values of the *K. obovate* assemblage were the highest, followed by the *S. apetala* and *S. caseolaris* assemblages. The AGC values of the *A. marina* assemblages were the lowest (Table 5). Overall, the average estimated AGC was 93.0 ± 24.3 Mg C ha$^{-1}$, which ranged from 31.7 Mg C ha$^{-1}$ to 195.8 Mg C ha$^{-1}$, while the total estimated AGC was 7903.2 Mg within the 85-ha mangrove forest (Table 5).

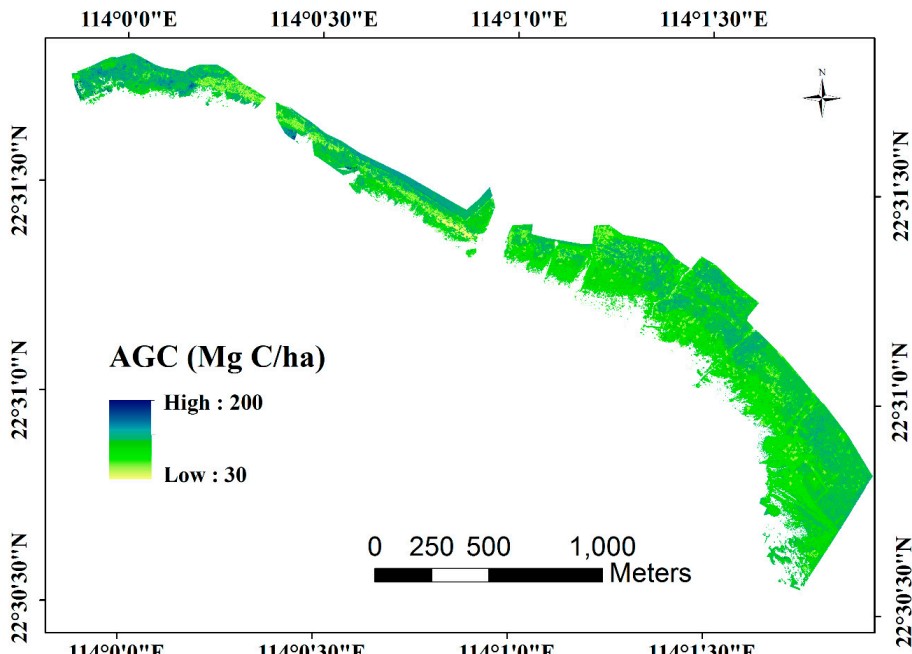

**Figure 7.** Aboveground carbon stock (AGC) values of the mangrove forest in the Futian Mangrove National Nature Reserve (FMNNR) based on an Random Forest model with 30 variables.

**Table 5.** Estimated aboveground carbon stock (AGC) values of four mangrove assemblages.

| Assemblage Types | Area (ha) | AGC (Mg C ha$^{-1}$) | Total carbon (Mg) |
|---|---|---|---|
| *A. marina* | 9.5 | 47.4 ± 6.2 | 450.3 |
| *K. obovata* | 62.6 | 94.0 ± 25.3 | 5884.4 |
| *S. apetala* | 7.6 | 128.6 ± 38.13 | 977.4 |
| *S. caseolaris* | 5.3 | 111.5 ± 31.7 | 591.1 |
| Total | 85.0 | | 7903.2 |

## 5. Discussion

### 5.1. The Importance of UAV Variables

Satisfactory results were achieved by estimating the AGC of the mangrove forests with all four types of variables (species, height metrics, VIs and texture features based on the RF algorithm in Experiment 1 (Table 4). The importance values (Figure 5a) and the results of Experiment 1 and Experiment 2 (Table 4) revealed that species type is the most important variable among all the selected variables for mangrove AGC estimation in this study, while Hmean was the second important. There were two main reasons why species was more important than Hmean for AGC estimation in this study. First, most of the mangrove forests are of similar age; thus, there was little variation in canopy structure for each mangrove assemblage from the CHM shown in Figure 3. Given a greater range of canopy height values, the CHM would have likely been more important in predicting AGC [36,37,39]. Second, mangrove species not only represented the wood density of each species, but also contained information related to structural characteristics such as tree density, DBH, and tree height, which were relevant to biomass estimation at the quadrat-level of different species. Furthermore, since each type of mangrove tends to have a similar canopy height, and it is a biophysical characteristic for estimating biomass in tree-level allometry, was nested within species [54,75,76]. Therefore, there were also some reports that species information plays a key role in biomass estimation modeling, and its inclusion can improve the estimation accuracy [16,21,23].

In this study, we also found that EXG and Var19 were important variables for estimating AGC (Figure 5). EXG values derived from RGB orthoimages had previously been shown to perform well in

the estimation of crop AGB in previous studies [34,67,77]. However, since visible spectral imagery lacks a near infrared band, many VIs are easily saturated, which will affect the subsequent AGC estimation of complex, dense forest stands [78]. As shown in Figure 4, NGBDI and NGRDI are totally saturated with the mangrove AGC. Meanwhile, Zhu et al. [21] found that red-edge band derived from the Worldview-2 is more sensitive than other multispectral bands to mangrove AGB.

On the other hand, a texture analysis, which refers to visual effects caused by spatial variations in tone quantity over a relatively small area, was adopted [79], because texture represents patterns in pixels that cannot be described by spectral values and individual pixels alone. Moreover, texture features show different canopy structures, which may reflect the tree density in mangrove assemblages. For example, the texture feature Variance represents the variance of the considered imagery. A high value indicated that the objects in the region were disordered, where large canopy gaps that reveal other objects such as water or mud may lead to lower mangrove AGC estimates. As the study showed, the integration of textural information with spectral information texture features yielded more accurate estimates compared to the use of spectral information alone [39].

## 5.2. Model Performance of the MLAs

The MLAs, which demonstrated great potential to estimate forest parameters, especially AGB and AGC, can overcome the multicollinearity problem and they do not make assumptions about the nature of the data distribution [44]. The comparison of the MLAs showed that RFs had better performance in AGC estimation than SVR and ANN in this study (Table 4). The RFs were more consistent in responding to small perturbations in the data, and the randomness in the RFs reduces overfitting during model training [71]. This result was consistent with some other studies that showed that RF and SVR have great potential for forest biomass estimation with remote-sensing techniques [41,44,46,47]. Additionally, the main advantage of the RF algorithm was to identify important predictor variables and model the relationship between them and the AGC. Therefore, the RF algorithm could provide a convenient and efficient way to estimate mangrove AGC with importance value measures.

## 5.3. Application and Limatation

To accurately estimate AGC values in the mangrove forest, we combined field sampling with UAV technology to obtain a high spatial resolution orthoimage and a DSM, which meant obtaining spectral and structural attributes simultaneously in the area. Additionally, the minimal time and labor required to cover the study area is one of the main advantages of the use of UAVs. The UAV took seven sorties of approximately 40 min each, including flight planning and scanning, with three people involved in the execution of the flights to cover the entire range of the study site (85 ha). In contrast, for the ground forest inventory, six people with experience in field sampling worked for one week (approximately 8 h per day) to obtain the ground data from an area of just 0.255 ha. The UAV also was able to access areas unreachable, or difficult to reach, on foot, particularly at low tides in areas with deep silt. Furthermore, the use of the UAV to monitor forests does not disturb the flora and fauna as much as traditional inventory surveys do.

In this study, we only used a simple digital camera loaded on the UAV, which has some limitations compared with hyperspectral cameras and laser scanning. In fact, more diverse spectral characteristics can be obtained when a hyperspectral camera is loaded on the UAV system, which can generate precise classifications of vegetation [26,80], and hence improve the biomass estimation [81]. Furthermore, the spectral saturation of dense canopies will be reduced as more VIs can be calculated and selected from the hyperspectral data [26,45]. In addition, to our knowledge, it is hard to obtain a sufficient number of SfM ground points in areas with highly dense canopies because the ground is not visible to the passive imaging sensor. In contrast, airborne laser scanning (ALS) can provide detailed structural variations along the canopy depth, while they can hardly provide sufficient spectral characteristics. With the development of remote sensors, UAV system with laser scanning and hyperspectral camera will be a very promising alternative in the estimation of forest AGB and AGC [82–84].

*5.4. Estimate of Mangrove Carbon Stocks*

The total mangrove C stock in the FMNNR was 7903.2 Mg over an area of 85 ha. The average predicted mangrove AGC was $93.0 \pm 24.3$ Mg C ha$^{-1}$, which is similar with the mean C density of $84.61 \pm 30.67$ Mg C ha$^{-1}$ in southern China [10]. Hutchison et al. [85] found a potential mangrove AGC of 67.96 Mg C ha$^{-1}$ in the FMNNR calculated with the latitudinal model, which is slightly lower than our estimate. The difference in the calculated AGCs may be due to the effort that the government implemented with regards to restoring the mangrove ecosystem [12]. As reported, the government started planting the exotic *S. apetala* and *S. caseolaris* in the FMNNR since 1993 [52], especially *S. apetala*, which is a fast-growing mangrove species from Bangladesh that has now become a dominant species in the FMNNR. Moreover, the *S. apetala* assemblage has a higher carbon density (Table 5) than other assemblages because of its high density, tall heights and large DBHs. As a result, there was a higher average C density than expected in the FMNNR. Thus, we suggest just conservation, rather than reforestation, as the next step for local management.

## 6. Conclusions

Our study effectively developed a remote-sensing model for the AGC mapping of mangrove forests using four kinds of variables, including species type, canopy height, texture features, and Vis, derived from a high spatial resolution orthoimage and DSM based on an UAV system with a digital camera in the FMNNR ($R^2 = 0.81$, rRMSE = 0.20, rMAE = 0.14). Moreover, the species-based model had better performance than the canopy-height-based model for AGC estimation, where mangrove species was the most important variable among those tested, with the mean height (Hmean) the second most important. Our results also suggested that the species-based model with VIs and texture features had acceptable performance for AGC estimation in the homogenous mangrove forests. Additionally, the RF algorithm had better performance in terms of mangrove AGC estimation than SVR and ANN in this study. Finally, the average predicted AGC was $93.0 \pm 24.3$ Mg C ha$^{-1}$, and the total estimated AGC was 7903.2 Mg for the mangrove forests. Based on the predicted data, we specifically suggest that conservation, and not reforestation, should be the next step in the local management of the mangrove forests in the FMNNR.

To conclude, species type information with spectral and structural attributes should be simultaneously considered when only simple RGB orthoimages are available for AGC estimation in areas of homogenous mangrove forests. In the future, the potential of applying UAV technology to determine C stocks and other biophysical parameters of mangrove forest should be investigated.

**Author Contributions:** Conceptualization, Z.L. and S.Y.; Methodology, Z.L., Y.C. and S.Y.; Software, Z.L. and Y.C.; Validation: Q.Z., Y.C. and S.Y.; Formal Analysis: Z.L.; Investigation, Z.L., Q.Y., and D.Z.; Resource, Q.Z and Q.Y.; Data Curation, Z.L., Q.Y., and D.Z.; Writing—Original Draft Preparation, Z.L.; Writing—Review & Editing, Q.Z. and S.Y.; Project Administration, Q.Z. and S.Y.; Funding acquisition, Q.Z.

**Funding:** This research was funded by the National Natural Science Foundation of China (Grant 31470513, 31770513) and a grant from Shenzhen City Administration Bureau.

**Acknowledgments:** We are grateful to many people who assist in the field survey.

**Conflicts of Interest:** The authors declare no conflict of interest.

## Appendix A

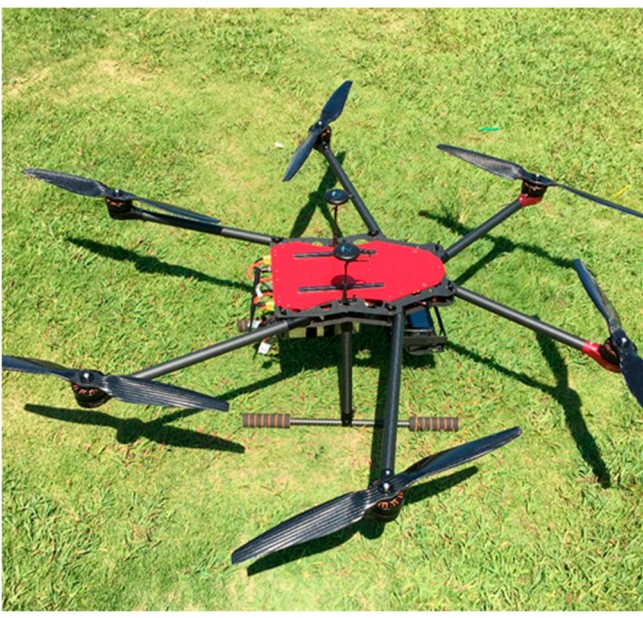

**Figure A1.** The six rotor-wing UAV system (ZR-66B, SOUTH, China).

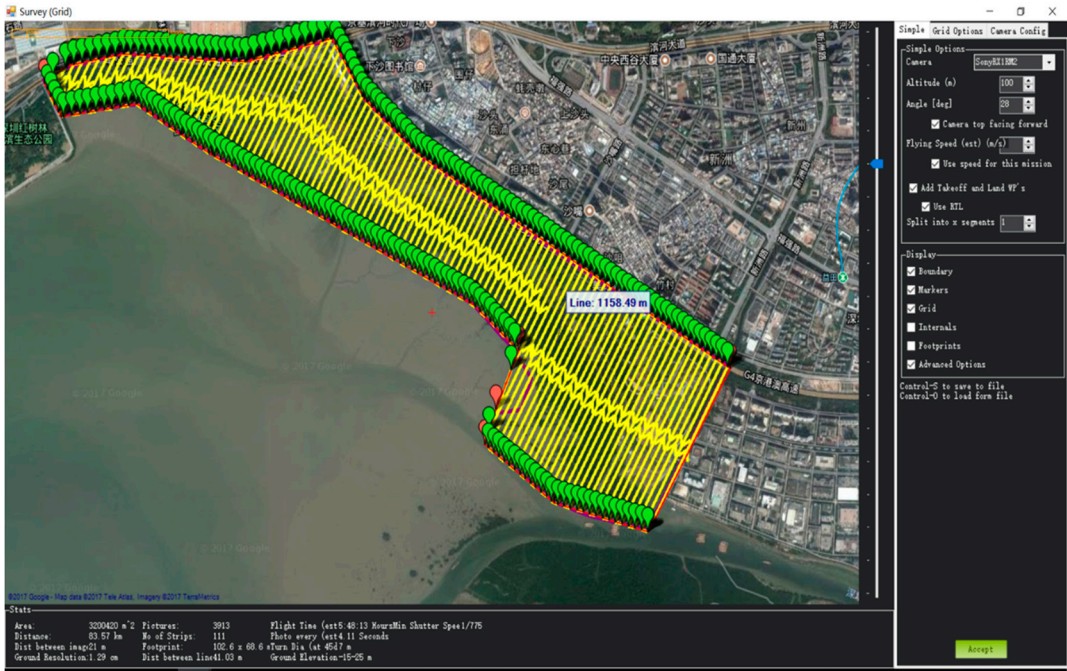

**Figure A2.** Flight Course Design.

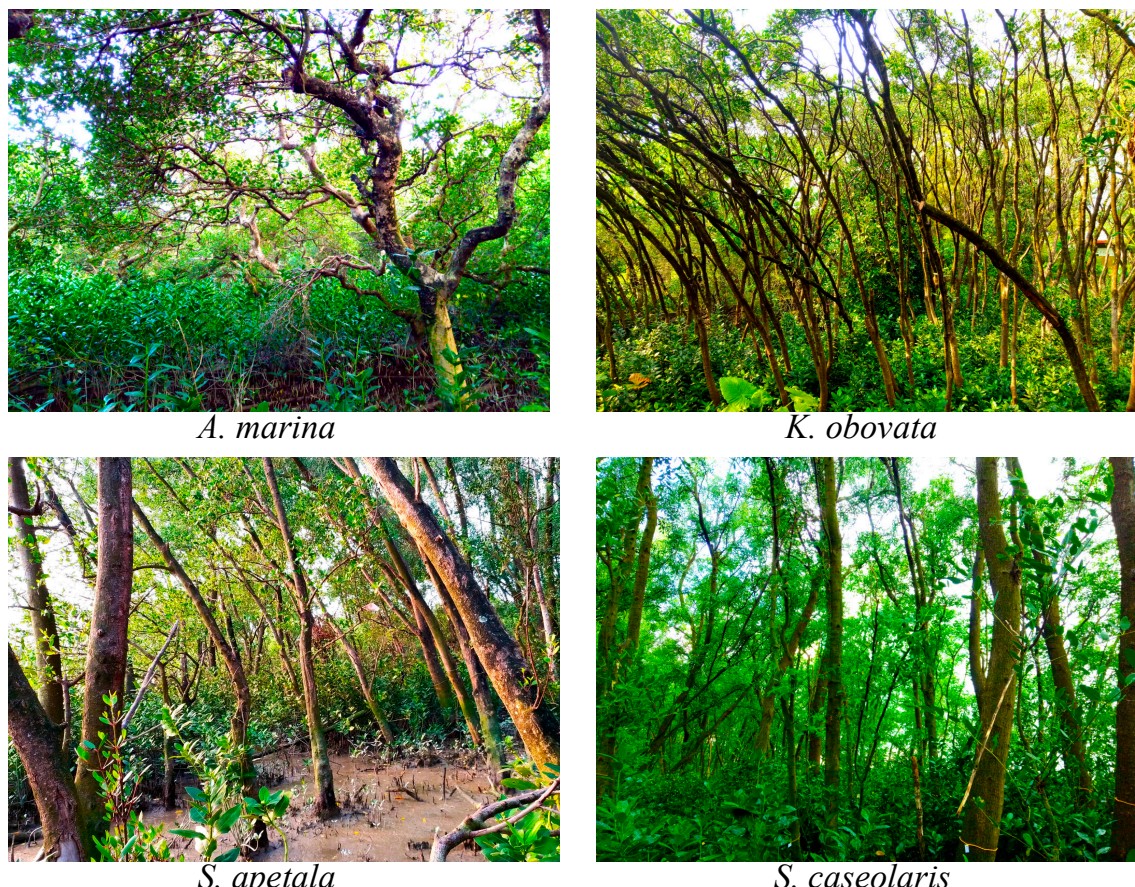

**Figure A3.** Photos of mangrove assemblages.

**Table A1.** Confusion matrix of the mangrove species classification for the UAV orthoimage.

| Mangrove Assemblages | A. marina | K. obovate | S. apetala | S. caseolaris | Producer's Accuracy | User's Accuracy |
|---|---|---|---|---|---|---|
| A. marina | 41 | 3 | 4 | 2 | 82% | 79% |
| K. obovate | 3 | 40 | 2 | 5 | 80% | 87% |
| S. apetala | 5 | 1 | 39 | 5 | 78% | 76% |
| S. caseolaris | 3 | 2 | 6 | 36 | 72% | 75% |
| Overall accuracy = 78% | | | | | | |
| Kappa coefficient = 0.73 | | | | | | |

**Table A2.** Biomass of each mangrove assemblage of field sampling.

| Assemblage Ttypes | Biomass (Mg ha$^{-1}$) | | | | |
|---|---|---|---|---|---|
| | Live Tree (stem) | Live Tree (branch) | Live Tree (leaf) | Dead Tree | Understory Vegetation |
| A. marina | 66.07 ± 1.99 [a] | 40.98 ± 0.55 [b] | 3.82 ± 0.78 [a] | 2.74 ± 1.57 [ns] | 2.50 ± 0.45 [a] |
| K. obovata | 213.59 ± 10.73 [b] | 12.32 ± 4.84 [a] | 4.97 ± 1.35 [ab] | 1.32 ± 0.45 [ns] | 8.48 ± 0.34 [c] |
| S. apetala | 190.78 ± 28.85 [b] | 46.67 ± 10.25 [b] | 5.78 ± 1.88 [ab] | 0 [ns] | 4.39 ± 0.61 [b] |
| S. caseolaris | 178.73 ± 18.15 [ab] | 38.11 ± 8.06 [b] | 5.11 ± 1.32 [ab] | 0 [ns] | 3.47 ± 0.81 [bc] |

Data is given as a mean ± standard error. The mean values of biomass followed by different letters (a,b,c) within columns are significantly different at $p < 0.05$ based on Dunn's tests, ns: non-significant difference.

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
