# Peer review of "Remote Estimation of Mangrove Aboveground Carbon Stock at the Species Level Using a Low-Cost Unmanned Aerial Vehicle System"

_remotesensing, doi:10.3390/rs11091018_

Round 1

Reviewer 1 Report

This is a re-review of the manuscript now entitled “Remote Estimation of Mangrove Aboveground Carbon Stock at the 2 Species Level Using an Unmanned Aerial Vehicle System.”  The manuscript has improved significantly from the first submission. Specifically, the addition of canopy height to estimates of AGC is key. I still have a few issues that need to be resolved before publication.

First, there is a tremendous level of additional text in this version of the manuscript. Excellent work. However, in most of the new text areas there are various grammatical issues that should be resolved, specifically words with incorrect tense. Please take some additional time to try to resolve these issues. Fixing these issues will enhance the readability of much of the manuscript.

Concerning more substantive material in the manuscript, I found the discussion of species vs canopy height importance skewed to preference species, when much of the literature has stated otherwise. Some of the statements like “species type is the most important variable among all the selected variables for mangrove AGC estimation” and “the height metrics derived from CHM will impart uncertainties into the estimated AGC values” are not qualified statements. Species was likely more important than canopy height in this study because there was almost no variation in canopy structure. This fact is not mentioned in the manuscript, as far as I can tell, but is apparent from the CHM shown in Figure 3. Given a greater range of canopy height values the CHM would have likely been more important in predicting AGC. Moreover, there is no evidence to suggest that the CHM adds additional uncertainty to the analysis. I’m curious why this was stated, when the results do not support this. They cite a range of crop studies showing height to be useful for predicting biomass of a single species, but mangroves are not crops. Height metrics are the most common way of estimating biomass in single-species and mixed species forests of every type (boreal, temperate, tropical) globally. The authors could improve the discussion by recognizing this and focusing more-so on why species was so important and not canopy height, rather than assuming canopy height is not important.

The key reason why canopy height was likely second to species in terms of importance is that, in this study, a large portion of the mangrove stand was planted and thus of very similar age. If you have a stand of trees that are all the same height, height will provide negligible explanatory power for a model predicting AGC. Species is a proxy for wood density, so if the stand is all the same height, and you only have variation in species, the variation in wood density will more directly capture spatial variability. This is the reason why estimates of average wood density are included in lidar-based estimates of tropical forest biomass. Otherwise, height alone would be too insensitive to changes in biomass or AGC.

My suggestion would be to focus the new portion of the discussion on this aspect and avoid saying canopy height is not important. In this study, the lack of canopy height variability is the primary reason why species is so important. Also, adding some citations from other mangrove or tropical forest AGC modeling, as opposed to crop studies, would be good.

Good luck on your revisions. I look forward to seeing the final version of this manuscript.

Author Response

Please see as attached

Reviewer 2 Report

Overall major comments

1.       Title and main text: The authors use throughout the text both the terms stock (AGC) and biomass (AGB). More consistency is needed.

2.       There are significant concerns for the approach used for extracting the CHM. The authors normalized the DSM to sea level. Bare earth should have been used for this task.

3.       The authors estimate the field measured carbon stock based on previous published results “The C mass of the vegetation  was calculated as the product of the vegetation biomass multiplied by the wood C concentration”.

To this end, they use 2 literature sources:

In the case of Ren et al. (2010), the C concentration rate for S. apetala (43%) is estimated considering also root biomass. Furthermore, the tree density in the study of Ren et al. is between 1328-1494 trees/ha, while in the current study is 1625 trees/ha.

In the case of Peng et al. (2016) vegetation carbon stock consisted of living trees (aboveground and belowground biomass), understory, pneumatophore, standing dead trees, fallen dead trees and litter

4.       The authors employ UAV based models of biomass estimation over mangrove forests. The best variable of their models is “species”. This variable is a forest species map, derived through photointerpretation of the UAV images. Field based AGB estimates of the authors are also species specific. Therefore, there is little scientific interest for the experiment considering the species map. Furthermore, this approach cannot be considered automated, given the interpretation stage included.

5.       The literature is missing some works recently published in this field (Pham et al. 2019, Navarro et al. 2019)

Specific comments:

Line 26 “all four types of variables”

Lines 68-70 IKONOS (or similar data) could fulfill the requirements for “fine scale” mapping

Line 73 “such an approach has the benefits of low cost and high flexibility.” Please support this statement by appropriate references

Line 83. This is not valid. UAV data has been used for forest biomass estimation

Line 101. This statement is contradictory with the one in line 108

Line 118 “added species type and canopy height”. Please rephrase

Line 220 “an artificial visual interpretation”. Not clear

Lines 228-231 The accuracy assessment results are not very reliable. Table S1, includes 88 records-that is the 5x5 plot sampled. That is a very small sample and not an appropriate sampling design. I would expect an independent random sample.

Section 4.1 should be re-allocated within Section 2.2

Lines 379-387, should not be included in results

Author Response

Please see as attached

Reviewer 3 Report

General comments: This paper deals with an interesting topic using an UAV for estimating aboveground carbon stock (AGC) at the species level. However, some aspects are still unclear that make it unsuitable for consideration for possible publication in Remote Sensing at the current form. Importantly, the authors did not calculate AGC of the shrub mangroves that may lead to the underestimation of the total AGC stock. Major revisions are required for the manuscript. Please do address all my concerns and comments in the revisions and resubmit the revised manuscript.

Specific comments:

+ Introduction: This section is lacking of the literature review as the follows: 

- Optical sensors multispectral and hyperspectral remotely sensed data for mangrove AGC estimation. 

Lines 78-82: Some examples stated in this manuscript are irrelevant. Why you mentioned the AGB of maize and aquatic plants? They are totally different from mangroves. 

- SAR sensors and LiDAR: various SAR data and LiDAR have widely used for estimating mangrove canopy heights and AGC in the tropics. 

- Remote Sensing approaches: various methods using linear and non-linear machine learning algorithms have been applied for AGB and AGC in mangrove ecosystems.

Please carefully read a recent review paper published in Remote Sensing and improve the literature 

" Remote Sensing approaches for monitoring mangrove species, structure, and biomass: Opportunities and challenges".

- Line 116: delete "are"

+ Materials:

- Why the authors did not calculate the shrub mangroves despite the fact that they are also significantly contributed  to the total AGC stocks because shrub mangrove are usually very dense. A recent paper published in Science highlighted the roles of small mangroves entitled: "the value of small mangrove patches"?

- Please add some real photos of different mangrove communities in the study area taken in the survey. 

- Table 1: The allometric equations used somehow are not so reliable as very low R-sq,. especially, for A. marina (0.45). Because the field surveys have done, the re-measurements are impossible. I suggest authors using the canopy diameter for some specific species ie. A. marina (Rsq= 0.97). Please see the paper below conducted also in the mangrove forests of China: https://www.sciencedirect.com/science/article/pii/S187802961100538X

- Field measurements data in December 2017 which may dry season; but the UAV flights were conducted in Aug 2017, which may rainy season. How authors solve this problem? Must explain in the revisions by providing the climate conditions (tidal conditions when conducting the survey in the study area and in the time of the UAV flights. And how to minimize effects?

-Lines 217-219: Various approaches can be used for mapping mangrove species, of which visual interpretation often is not superior compared to object-based approaches using very high spatial resolution. Please rephrase this statement by reading suggested review paper above.

As shown in Fig. 3, some species are mixed together and this situations are very common in the mangrove ecosystems. But, authors did not take into account for classifying of mixed species, that also very challenging topics in Remote Sensing. 

+ Model regression and accuracy assessment

- More clarifications and citations are required for example in Lines 293-298.

Line 362: Please correct "As shown in Table 4"

- Table 5: "K. obovata" not "obovate".

+ Authors have to discuss more the data saturation for the use of UAV for estimating AGC for mangrove species level by providing a AGC range and ML models performance compared to other similar studies using various remote sensing data.

+ Please Correct section 5.3. Applications and Limitations

- Authors must discuss the main limitation caused by choosing the allometric equations used in this study and ignoring shrub mangroves.

Author Response

Please see as attached

Round 2

Reviewer 2 Report

The authors addressed all the concerns

Reviewer 3 Report

Thanks for providing the revised version and the authors have addressed my comments and suggestions when revising this. Therefore, the quality has improved compared to previous one.

The final version should be edited by a native English speaker or professional editing service.

This manuscript is a resubmission of an earlier submission. The following is a list of the peer review reports and author responses from that submission.

Round 1

Reviewer 1 Report

The study used drone-based orthoimages to classify species and estimate aboveground carbon (AGC) storage. Overall, the work is scientifically sound and the manuscript is well written. I have one major suggestion that would improve the utility of the findings and put the work more in the context of the current state of AGC estimation.

The main issue I have is that the drone was only used to create an orthoimage, rather than a canopy model. The study is based in an inherently flat environment, where canopy height models with structure-from-motion is simplest. The only need is to normalize to the sea-level and a canopy height model would be available to this study.

The reason why this matters is because your final RF model may be inflating the importance of species, especially without considering a biophysical variable such as height. The texture metrics are no doubt useful in improving the final model and differentiating species and height of species, but why not simply use the canopy height model?

I imagine incorporating canopy height into the RF model would show canopy height as being the most important predictor of AGC storage with species second. This creates a more physically-based predictive model, relying on the species – essentially wood density – and height, both variables that are already included in tree-level allometry.

Another valuable change to the study would be to test your estimates of AGC based on the RF model only using canopy height and only using the species-based model. If the species and textural indices are better than canopy height alone that would be a very powerful finding. Otherwise, it is difficult to determine what the baseline estimates would actually be.

Finally, please report bias (relative and absolute) in the model assessment. The addition of this statistic says a lot about how well the model performs.

Overall, the work is good, but a base of comparison through the inclusion of canopy height (which is easily acquirable with the current dataset) would make this a truly valuable contribution.

Minor:

Line 39: Mention how carbon dense mangroves are. These numbers are more tangible and offer a direct comparison to typical carbon densities.

Reviewer 2 Report

Overview:

This study focuses on estimating aboveground carbon (AGC) of mangrove species using vegetation indices and texture features that derive from high resolution RGB UAV images.

In general, the paper is not-properly written and structured. The methodological design is weak and require extensive improvement.

General comments on paper:

The introduction should be improved by including studies based on UAV and VHR/data for AGC estimation. References should be also included in relation to the approaches/algorithms used. Finally, references on the use of texture features in AGC estimation or relevant studies are further missing. For example, reference [55] could be mentioned.

Comments by line:

        78           UAVs do not acquire aerial photographs

        84-89     The aim and the objectives should be re-written

                    86           typo in “ABG” (should be “AGB”)

        In Fig. 1 it is suggested to provide a grid with measuring units on the species plots. Also, increase font size of the various elements included within Fig.

Fig. 1 Location

        116         What is the crown width of the species measured? It seems that plot size is extremely small. How did a handheld laser range finder was used within such dense stands as the ones depicted within Fig, 1?

        116-117                How monodominance was determined beforehand? How this was done in a random manner?

121         sentence “In each quadrat…” not so clear

130         Wood density usage is not clear. The abstract of the reference study provided (i.e. 46) does not indicate the usage/development of an allometric equation since the focus seems to be on LULC mapping.

158         a UAV collects an aerial image, not an orthoimage. In later analysis the image is orthorectified.

Section 2.4 What is the accuracy of the rectification process

160         What kind of GPS?

162        There are far more appropriate spectral bands for vegetation detection and carbon mapping

176         Not valid. There are many studies based on automatic or semi-automatic classification

177         artificial visual interpretation??

187         “statistical models” or “ensemble learning methods” maybe?

            207         maybe 25 x 25 instead of 250 x 250? This result to a significant generalization of the information

            210         There is no justification for the of the use of a  3 x 3 kernel size (Table 2).

Section 2.1 This part belongs to “materials and methods”

232         where are the allometric equations in Table 2? Maybe “were calculated using allometric equations from the variables in Table 2”?

236         in Table 3, what does a,b,c stand for in the significantly different values?

242         The authors should not provide accuracy measures for a visual interpretation procedure. They should rather improve the mangrove distribution map, since this is the intermediate data of the research

243         there is no Table S4 available in the paper

243         how many and which samples were used for the classification accuracy assessment?

        In Section 2.5. It is not clear how the classification was performed. What method was used? Where the species delineated by the experienced researchers? If so, this is not very clear in the text and also in Fig. 2, the classification seems to be done at a pixel level.

        304         Within study (18) there is no comparative assessment to justify that “species type was the most important variable among all the input parameters”. Not sure about the other two studies cited

306         sentence “Since RGB…” not so clear and is maybe contradictory.  

327         Maybe the reference [69] should be put next to “Hutchison et al.”?

365         other word than “explained”?

Reviewer 3 Report

General comments: The authors present an interesting work. However, the current version has lots of weaknesses and shortcomings make it unsuitable for consideration for possible publication in Remote Sensing. The current paper is lack of literature review on remote sensing approaches for estimating mangrove AGC and classifying mangrove species tasks. I found that the citations are inconsistent with reference lists. The authors only investigated the mangrove AGB for the mangrove trees with DBH  ≥ 5 cm could mislead the AGC for mangrove species. I believe all live mangrove trees should be measured using different allometric equations. Methods used are still unclear and not enough explanation. It is not clear how authors used for the training and the validation sets considering two tasks: mangrove species classification and AGC regression models. Authors should present the research flowchart or framework how they can estimate AGC for different species.

Specific comments:

- Introduction:

+ This part is lacking literature review on remote sensing-based approaches for mangrove AGC estimation including parametric and non-parametric regression models.

+ As authors also considered the mangrove species discrimination task, the literature review for this also need to be taken into account.

- Materials and methods:

+ This part is still unclear. Authors have to show the research flowchart consists of two tasks:

Tree species discrimination and their AGC estimation.

+ How many samples and Groundtruth used for training and validation sets considering two tasks: mangrove species classification and AGC regression models? More explanation is required regarding the classification and regression tasks.

+ It is clear to say that K. obovata and S. caseolaris are normally planted together to protect the sea dykes. As a result, the mixed species of several mangrove communities are relatively common in the tropics. How the authors handle this issue?

+ Why authors only test the RF models for both classification and the AGC estimation?

Authors should compare several well-known machine learning algorithms for two tasks

+ Authors should present the scatterplots between each variable and measured-AGB for different species to make the readers understand the data saturation for the use of UAV orthoimages in the current work.

- Discussion:

+ Authors must discuss the current limitations for the use of UAV orthoimages for the AGC estimation and the limitations of tree species discrimination in terms of overall accuracies and Kappa coefficient.

I would like to see the data saturation and authors must discuss this issue more thoughtfully.